# Roles of DNA Methylation in Color Alternation of Eastern Honey Bees (*Apis cerana*) Induced by the Royal Jelly of Western Honey Bees (*Apis mellifera*)

**DOI:** 10.3390/ijms25063368

**Published:** 2024-03-16

**Authors:** Amal Abdelmawla, Xin Li, Wenkai Shi, Yunlin Zheng, Zhijiang Zeng, Xujiang He

**Affiliations:** 1Honey Bee Research Institute, Jiangxi Agricultural University, Nanchang 330045, China; aaa28@fayoum.edu.eg (A.A.); 17852099307@163.com (X.L.); swk18792464186@163.com (W.S.); zhengyl8011@163.com (Y.Z.); 2Jiangxi Key Laboratory of Honey Bee Biology and Bee Keeping, Nanchang 330045, China

**Keywords:** honey bees, nutritional crossbreed, body color alteration, DNA methylation

## Abstract

Honey bees have a very interesting phenomenon where the larval diets of two different honey bee species are exchanged, resulting in altered phenotypes, namely, a honey bee nutritional crossbreed. This is a classical epigenetic process, but its underlying mechanisms remain unclear. This study aims to investigate the contribution of DNA methylation to the phenotypic alternation of a *Apis mellifera–Apis cerana* nutritional crossbreed. We used a full nutritional crossbreed technique to rear *A. cerana* queens by feeding their larvae with *A. mellifera* royal-jelly-based diets in an incubator. Subsequently, we compared genome-wide methylation sequencing, body color, GC ratio, and the DMRs between the nutritional crossbreed, *A. cerana* queens (NQs), and control, *A. cerana* queens (CQs). Our results showed that the NQ’s body color shifted to yellow compared to the black control queens. Genome methylation sequencing revealed that NQs had a much higher ratio of mCG than that of CQs. A total of 1020 DMGs were identified, of which 20 DMGs were enriched into key pathways for melanin synthesis, including tryptophan, tyrosine, dopamine, and phenylalanine KEGG pathways. Three key differentially methylated genes [*OGDH*, *ALDH(NAD+)* and *ALDH7*] showed a clear, altered DNA methylation in multiple CpG islands in NQs compared to CQs. Consequently, these findings revealed that DNA methylation participates in *A. cerana–A. mellifera* nutritional crossbreeding as an important epigenetic modification. This study serves as a model of cross-kingdom epigenetic mechanisms in insect body color induced by environmental factors.

## 1. Introduction

Epigenetics precisely regulates gene expression through DNA methylation, histone changes, non-coding RNAs, and chromatin remodeling [1,2]. Epigenetic processes, such as histone post-translational modifications, DNA methylation, and non-coding RNAs can rapidly and flexibly change transcriptional pathways that produce different phenotypes [3]. Honey bees are a keystone species for epigenetic studies, since they have many common epigenetic phenomena, such as caste differentiation. Another classic epigenetic phenomenom is called nutritional crossbreeding, where larval diets are exchanged between two honey bee species, resulting in different phenotypes. This phenomenon has been frequently reported in Asian and European honey bees. Studies have accumulated evidence that nutritional crossbreeds can be caused by alternations in phenotypes, including changes in body color, morphology, and behavior [4,5]. Specifically, previous studies reported notable body color changes in nutrient–hybrid *A. cerana* and *A. mellifera* adults [6,7,8]. For the *A. cerana–A. mellifera* nutritional crossbreed, prior studies have extensively researched the crucial roles of partial nutritional crossbreeding and its essential roles in the epigenetic mechanism. Xie et al. reported decreases in wing size, proboscis length, and the lengths of the third and fourth dorsal plate of the abdomen and altered the mite-defending ability of *A. mellifera* workers by nutritional crossbreed [8]. Feeding with a *A. mellifera* larval diet also increases the body size, birth weight, and ovariole numbers of *A. cerana* workers [9,10]. Shi et al. showed that a *A. cerana–A. mellifera* nutritional crossbreed could alter the genome-wide alternative splicing of *A. cerana* [11]. Recently, our previous findings revealed that LncRNAs and miRNAs play a key role in the full nutritional crossbreed of *A. cerana* and *A. mellifera* by regulating the expression of key genes of melanin biosynthesis pathways resulting in an alternation in the body color of *A. cerana* queens [12]. These findings emphasize that body color alternation is one of the most noticeable occurrences resulting from *A. cerana–A. mellifera* nutritional crossbreeding.

The honey bee methylome is sensitive to the nutrition of the development larvae, so an early nutritional environment establishes the larva on either a worker or queen developmental pathway [13]. Since the methylation of the honey bee genome is selectively altered by the nutrient inputs/dietary environment during development, resulting in distinctly different phenotypes of the worker and queen bees [13,14,15]. DNA methylation is an epigenetic modification primarily responsible for individual phenotypic variations. This modification has been reported to play an important role in the caste differentiation, brain plasticity, and body development of honey bees [16]. For instance, differences in queen and worker larval diets induce changes in DNA methylation contributing to queen–worker dimorphism [13,14,15,17,18]. In addition, a genome-wide association has been reported among differential gene expression, splicing, and differential DNA methylation the overlapped GO terms point to nutrition-related activity in primitively eusocial *Bombus terrestris.* Furthermore, DNA methylation plays an important role in regulating honey bee queen development. He et al. (2021) have proved that differentially methylated genes (DMGs) induced by maternal effects are strongly related to queen–worker differentiation, indicating that the honey bee maternal effect could cause DNA methylation in queen rearing and potentially contribute to queen development [18]. The methylomes of queens reared from eggs and worker larvae differed over four generations, accumulating increasing differences in exons of genes related to caste differentiation, growth, and immunity [17]. Consequently, DNA methylation induced by honey bee larval diets, can function in honey bee phenotypic alternation, reproduction, longevity, immunity, and metabolic [19]. Here, we hypothesized that the honey bee’s nutritional crossbreed induced by different larval diets from distinct honey bee species would be possibly controlled by epigenetic factors. Consequently, DNA methylation, as non-coding RNAs [12], would also contribute to the regulations of phenotypic alternation in *A. cerana–A. mellifera* nutritional crossbreed.

Previous studies have shown body color alternation in various animals, including birds [20], fish [21], vertebrates, mice [22,23], and mammals [24], revealing that DNA methylation could alter body color. Zhang et al. (2017) showed that the molecular mechanism of body color variation in *Crucian carp* is strongly related to disruptions in gene expression and DNA methylation during pigmentation [21]. Additionally, Chen et al.’s (2021) findings added to our understanding of the epigenetic regulation of rabbit pigmentation by demonstrating a link between inherited color dilution and DNA methylation changes in hair follicles [24]. More recent research by Dolinoy et al. (2006) and Duncan et al. (2022) revealed a connection between epigenetics and phenotypic plasticity, with coat color varying from yellow to pseudoagouti due to the variable DNA methylation of a transposable element in the agouti coat color gene locus, which can be altered by the maternal diet [22,25]. However, it is unclear whether DNA methylation participates in the regulation of body color alternation in insects.

Previous evidence has shown that exchanging the larval diets between *A. cerana* and *A. mellifera* results in a clear body color alternation [7,8,12,26], and some non-coding RNAs are involved in this color alternation via regulating the expression of key genes which are related to the melanin synthesis [12]. However, it is still unclear whether other epigenetic modifications such as DNA methylation are involved in the regulation of those key genes. Functionally, DNA methylation is associated with the regulation of gene expression and the silencing of transposable elements [27]. In honey bees, DNA methylation is a vital epigenetic modification participating in caste differentiation [28], the division of labor, and behavioral plasticity [29,30]. 

However, genome-wide DNA methylation involved in the body color regulation of insects and honey bees is not yet fully understood. Therefore, the aims of our study were to investigate whether DNA methylation is associated with body color regulation in *A. cerana–A. mellifera* nutritional crossbreeding through genome-wide methylation sequencing, providing a foundation for further exploration of the specific role of DNA methylation and candidate genes in body color alternation. 

## 2. Results

### 2.1. Overview of Whole-Genome Bisulfite Sequencing and DNA Methylation

A total of 44,645,929,800 clean data (bp) reads were obtained from six libraries. After filtering, the Q20 (%) values of each sample were over 90% (Table 1 and Appendix A), indicating considerably high sequencing quality. The six libraries were used to generate genomic DNA and analyzed based on whole-genome bisulfite sequencing (WGBS), resulting in an average of 7,648,947,800 bp for NQs and 7,233,028,800 bp for CQs. Detailed results of the methylation sequencing data are shown in Table 1. For NQs, the average number of clean reads was 50,992,985.33 and for CQs it was 48,220,192, with high quality clean reads being no less than 99% for each sample (Appendix A). Pearson’s correlation coefficient of all samples and biological replicates was above 0.8 (Appendix A), indicating a high repeatability.

The sequencing depth per sample compared to the reference genome was more than 20× per strand for each sample. The reference genome mapping rate ranged from 69.43 to 72.88%, indicating that the data could be available for subsequent analysis (Appendix A). The average methylation level of the whole genome was 0.83% of genomic cytosines 1.30 and 0.58, and 0.68% of these were CG, CHG, and CHH types, respectively (Appendix A).

Pie charts demonstrated the proportions of mCG, mCHG, and mCHH between NQs and CQs, with mCG being the most abundant, ranging from 48.31 to 69.77% (Figure 1A). Our results also indicate that NQs had a much higher percentage of mCG compared to CQs (Figure 1A), suggesting that DNA methylation possibly plays an important role in honey bee nutritional crossbreeding. The CPG methylation clustering tree exhibits distinct differences between samples of NQs marked in blue and CQs marked in red (Figure 1B).

Our data show the distribution of methylation levels in different genomic functional regions through the CG sites of NQ/CQ comparison (Appendix A). The DNA methylation level in both NQs and CQs dramatically decreased in CHG and CHH, while the mCG/CpG islands demonstrated the highest methylation levels, 0.05 bp for body genes and genomes downstream. Interestingly, the methylation levels in the upstream region dramatically decreased (Appendix A).

### 2.2. DMRs and DMGs between NQs and CQs

Overall, we identified 1436 DMRs corresponding to 1020 DMGs (Figure 2A,B, Appendix A). In total, 472 DMGs were up-methylated, and 548 DMGs were down-methylated (Figure 2B). The top 100 DMR clustering (heat map) showed a clear difference between NQs and CQs (Figure 2C).

### 2.3. GO and KEGG Enrichment

The enrichment and classification of GO terms are summarized in Appendix A. The top four GO terms of DMGs were closely related to body color regulation, which were as follows: GO:0043227 membrane-bounded organelle, GO:0005622 intracellular, GO:0043231 organelle, and GO:0044444 cytoplasmic part.

In total, DMGs were enriched into 274 KEGG pathways (Appendix A), in which the top 20 significant pathways included the mRNA surveillance pathway, inositol phosphate metabolism, phosphatidylinositol signaling system, homologous recombination, and mTOR signaling pathway, as shown in Figure 3A and Appendix A. These top 20 pathways included amino acid metabolism (Tryptophan) and environmental information processing (mTOR, MAPK, Notch, and Wnt signaling), which are involved in regulating queen development and organismal systems [31]. Interestingly, the tryptophan pathway formed an important KEGG pathway controlling body color (Figure 3B), as well as dopamine and the biosynthesis of amino acids pathway [32,33,34,35,36].

### 2.4. DMGs Related to Body Color Regulation

Figure 4E clearly showed different body colors between NQs and CQs. Of the 1020 DMGs, 20 DMGs were enriched into four key pathways (tryptophan, tyrosine, dopamine and phenylalanine; see Figure 3B and Appendix A), which are key pathways for honey bee body color formation. We compared the DMGs with DEGs from our previous study [12]. Interestingly, three DMGs were the same as the DEGs (*ALDH7A1*, *ALDH2*, *OGDH*) that are enriched in the tryptophan pathway. The tryptophan pathway is vital for the formation of honey bee body color, resulting in an alternation from black in CQ control queen to yellow in NQ queens [12]. Our data clearly show that the various methylation sites of these three key DMGs [*OGDH*, *ALDH(NAD+)* and *ALDH7*] differed between NQs and CQs, specifically those marked in red sites in (Figure 4A–C), reflecting the notable changes in DNA methylation of those key genes induced by *A. cerana–A. mellifera* nutritional crossbreeding. Therefore, these findings indicate that DNA methylation contributes to body color alternation by being involved in the key pathways in the melanin synthesis process.

## 3. Discussion

The honey bee nutritional crossbreed is a classic epigenetic process, and the body color alternation is a typical trait of this phenomenon. Yet, the potential role of genome-wide DNA methylation involved in the body color regulation of insects has remained a mystery.

Our previous study reported that the body color of nutritionally crossbred *A. cerana* queens shifted from black to yellow by feeding *A. cerana* larvae with *A. mellifera* royal jelly, and these phenotype changes are mediated by regulating key the expression of DEGs in melanin biosynthesis pathways, such as *DDC*, *TH*, *OCGH*, *ALDH*, *KMO*, and *TDC* [12]. In the present study, body color alternations in honey bee queens were induced by *A. cerana*–*A. mellifera* nutritional crossbreeding (Figure 4E), which is consistent with previous studies [6,7,8,12]. Insect integuments contain key pigments like melanins, which are derived from catecholamines such as dopamine. These catecholamines are produced by tyrosine-mediated cuticle tanning metabolism, essentially contributing to the darkening and hardening insect cuticles. The melanin precursors (e.g., dopamine) are synthesized in epidermal cells, the resulting catecholamines are secreted into the cuticle, where they are further oxidized to melanin pigments [37]. Insect melanins are secreted into cuticles, and dark-colored melanins are predominantly derived from dopamine, whereas light-colored melanins are mostly derived from N-β-alanyldopamine (NBAD) and/or N-acetyldopamine (NADA) [38]. The major sources of the pigments are aromatic amino acids (i.e., tyrosine, phenylalanine, and tryptophan). Tyrosine is converted to dopamine by *TH* and *DDC.* Subsequently, dopamine is converted to NBAD and NADA by NBAD synthase and arylalkylamine N-acetyltransferase (aaNAT), respectively. In addition, NBAD can be converted to dopamine and β-alanine by NBAD hydrolase (NBADH), which catalyzes the reverse reaction of NBADS [38,39,40,41,42]. Therefore, genes enriched into key pathways, including tryptophan, tyrosine, dopamine, and phenylalanine, that are involved in melanin synthesis are important for insect body color formation. Our data show that an amino acid metabolism pathway (tryptophan pathway) was identified in the top 20 significantly enriched pathways between NQs and CQs (Appendix A). Twenty key genes which were involved in four key pathways for melanin synthesis were also significantly differentially methylated between NQs and CQs (Figure 3).

Many studies argued that the body color of bees is a stable ecological trait [43,44]. Genes such as yellow and ebony genes determine the patterns and intensity of melanization, which contribute to the insect body color formation [45]. Knocking out the yellow-y gene induced a reduction in the amount of black pigment in worker bees’ cuticles, while the expression of Amyellow-y and aaNATA in mutant drones, which have a dramatic body pigmentation defect, was lower than in wild-type drones [46]. Previous research indicated that the ebony gene contributes to the diverse body color variation of honey bees, suggesting that genes like ebony might be evolutionarily selected and vary in different bee species or subspecies resulting in distinct body colors [47]. Therefore, the regulation of body color in honey bees not only depends on the key pathways for melanin synthesis but can also be influenced by other conserved body-color-related genes. In this study, a clear alternation of body color was observed in NQs. This color change is an environmentally induced epigenetic process. Interestingly, we did not identify genes like ebony, yellow, or yellow-y, which were differentially expressed in our previous study [12] or methylated in the present study, but identified other key genes for melanin synthesis. Perhaps these genes are more active compared to genes like yellow and could be regulated by epigenetic modifications, which still requires further investigations.

Moreover, this phenotypic alternation is a classical epigenetic model which is induced by nutritional diets; therefore, the epigenetic modifications play a key role in this process. As one of epigenetic modifications, DNA methylation plays an important role in the regulation of gene expression in insects [48]. In the present study, our genomic DNA methylation results also showed a dramatic difference between NQs and CQs, with a higher methylated mCG in NQs compared to CQs (Figure 1 and Figure 2), with 1436 and 1020 DMRs and DMGs between them, respectively (Figure 2). Twenty key DMGs were enriched into four key KEGG pathways (tryptophan, tyrosine, dopamine, and phenylalanine). In particular, three key DMGs (*OGDH*, *ALDH7A1*, and *ALDH2)* are highly involved in melanin biosynthesis, which is also identified in our previous study, while the other 17 DMGs remain under analysis for further confirmation. Clear differences for each CpG island of the three DMGs [*OGDH*, *ALDH(NAD+)* and *ALDH7*] are shown in Figure 4A–C, reflecting a good example of DNA methylation regulating the alteration of body color from black in CQ control queens to yellow in NQ queens, possibly via altering the expression of key genes. Therefore, these findings clearly show that DNA methylation plays an essential role in the body color alternation of *A. cerana–A. mellifera* nutritional crossbreeding. Additionally, previous studies show that the mCG type is the most important type among three DNA methylation types in honey bees and dominate the functions of DNA methylation in honey bee caste differentiation, behavioral plasticity, etc. [16]. Here, the mCG type was also the main DNA methylation type/pattern in honey bee nutritional crossbreeding, suggesting that these mCG-type methylation alternations possibly play a key role in the regulations of phenotypic plasticity in *A. cerana–A. mellifera* nutritional crossbreeding. 

There is increasing evidence that DNA methylation plays a key role in the alteration of body color in various animals, including birds, fish, mammals, and vertebrates. Recent findings by Chen et al. (2021), have expanded our understanding of how epigenetic mechanisms control pigmentation in rabbits through demonstrating a correlation between inherited color dilution and changes in DNA methylation in hair follicles [24]. Similarly, Zhang et al. (2017) revealed a connection between gene expression and DNA methylation disruptions during *Crucian carp* pigmentation [21]. Maternal diet can influence DNA methylation and coat color in mice [22]. Our results are consistent with previous findings, whereas a clear alternation of DNA methylation is induced by *A. cerana–A. mellifera* nutritional crossbreeding. Moreover, our data show that the tryptophan pathway, which is a KEGG pathway for controlling the pigmentation of queen bodies, was one of top 20 significantly enriched KEGG pathways, and 31 key genes involved in melanin synthesis were significantly methylated. Therefore, it is believed that DNA methylation plays an important role in the body color alternation in honey bees via regulating the expression of key genes that are involved in melanin synthesis.

Here, we caution that DNA methylation is not the unique epigenetic modification that is involved in *A. cerana–A. mellifera* nutritional crossbreeding. Our previous study has already shown that eight non-coding RNAs also contribute to regulating the expression of key DEGs that are enriched in three key pathways for melanin synthesis [12]. Royal jelly is a very powerful food for the developmental regulation of honey bees. For instance, various studies showed that multiple omics such as DNA methylation, non-coding RNAs, histone modifications, 3D chromosome structure, and m6A methylation all participate in the determination of queen–worker honey bee differentiation [15,49,50,51,52]. Our recent study has shown that honey bee caste differentiation is regulated by an interacting system of many epigenetic modifications [52]. In this study, the phenotype plasticity of *A. cerana–A. mellifera* nutritional crossbreeding is also induced by honey bee royal jelly. Furthermore, DNA methylation and non-coding RNAs [12] both contribute to the body color alternation of *A. cerana–A. mellifera* nutritional crossbreeding. Consequently, other epigenetic modifications, such as histone modification and 3D chromosome structure, may also join with DNA methylation and non-coding RNAs to regulate the body color alternation of *A. cerana–A. mellifera* nutritional crossbreeding.

In summary, this study firstly investigated the epigenetic modification of DNA methylation in body color alternation in honey bee nutritional crossbreeding processes. It is clear that the honey bee nutritional crossbreeding induces DNA methylation changes in NQs and subsequently participates in the regulation of the expression of key genes for melanin synthesis, affecting their pigmentation. Our findings, for the first time, revealed that DNA methylation is strongly involved in the body color alternation of *A. cerana–A. mellifera* nutritional crossbreeds. This study serves as a model of phenotypic plasticity induced by nutritional stimuli and its epigenetic mechanisms to other animals and plants.

## 4. Materials and Methods

### 4.1. Queen Rearing and Sampling

The Institutional Animal Care and Use Committee of Jiangxi Agricultural University, China, approved all experimental procedures using honey bees in our study with an approval number of jxaull-2021-25. In total, nine colonies of *Apis cerana* and *Apis mellifera* were maintained at the Honey Bee Research Institute, Jiangxi Agricultural University, Nanchang, China.

Six healthy *A. cerana* colonies were used as one-day-old larvae suppliers to artificially rear nutritional crossbreed queens (NQs), and same larvae were reared in three of these six colonies to produce natural queens as a control (CQs). Each *A. cerana* colony had a mated queen and around 12,000 worker bees. Three strong *A. mellifera* colonies were used for fresh royal jelly collection, and each colony had a mated queen and more than 30,000 worker bees. In total, six *A. cerana* colonies and three *A. mellifera* colonies were used in this study.

### 4.2. A. cerana Nutrient Crossbred Queen Rearing

To artificially rear *A. cerana* nutrient crossbred queens, a honey bee larvae diet formula was prepared according to our pervious study [12]. The diet formula was as follows: 53% fresh A. mellifera royal jelly (AmRJ), 6% D-glucose (purity: analytical reagent, Xilong Scientific, Shantou, China), 6% D-fructose (purity: ≥99%, Solarbio life sciences, Beijing, China), 1% yeast extract (Lot: 2194133, Oxoid Ltd., Hants, UK), and 34% distilled autoclaved water. The diet was created using sterile laboratory tools and beakers, and all diet components were warmed up to room temperature prior to mixing, using a spatula [53].

To obtain young *A. cerana* larvae, *A. cerana* queens were caged on an empty comb to lay the same-age eggs for 6 h, according to [12]. The queens were released and combs with eggs were placed into a queenless area of the hive. Similarly, half of the newly hatched *A. cerana* larvae (6 h) were transplanted into 24-cell tissue culture plates, and then incubated at 35 °C and 95 ± 3% relative humidity (RH). Each cell with one larva was primed with a 200–400 µL food formula, which was increased daily according to the larval age. These larvae were fed for six days and then entered the pupation stage. For pupation, the 6-day-old larvae were transferred to 12-cell tissue culture plates lined with a piece of Kimwipe and kept in an incubator at 35 °C and 80% RH for 7 days. Fully developed queens, with notches in their mandibles and at least 16 days of development, were sampled. 

### 4.3. Control Queen Rearing

Correspondingly, for the control group, three biological replicates of the remaining hatched *A. cerana* larvae were transplanted into wax queen cells to obtain hive-reared queens in their native queen-less colonies. Each replicate had 30 one-day-old *A. cerana* larvae. After emergence, six fully developed NQs and six CQs were flash frozen in liquid nitrogen and stored in a −80 °C freezer for further genomic methylation analysis.

### 4.4. Genome-Wide Methylation Analysis

Each group consisted of three biological replicates, with two queens in each biological replicate used for genome-wide methylation testing. The whole bodies of the queens were used for genomic DNA extraction using the StarSpin Animal DNA Kit (GenStar, Beijing, China) according to manufacturer’s introductions. 

### 4.5. Library Construction and Sequencing

After the extraction of genomic DNAs from the samples, their DNA concentration and integrity were detected using a NanoPhotometer^®^ spectrophotometer (IMPLEN, Westlake Village, CA, USA) and Agarose Gel Electrophoresis, respectively. We also prepared DNA libraries for bisulfite sequencing. Briefly, the genomic DNAs were fragmented into 100–300 bp reads by Sonication (Covaris, Woburn, MA, USA), and then purified with MiniElute PCR Purification Kit (QIAGEN, Germanown, MD, USA). The genomic fragments were ligated to methylated sequencing adapters, and the end repaired. A single “A” nucleotide was added to the 3′ end of the blunt fragments. Then, the genomic fragments were ligated to methylated sequencing adapters. Fragments with adapters were bisulfite converted using a Methylation-Gold kit (ZYMO, Orange County, CA, USA), and unmethylated cytosine was converted to uracil during sodium bisulfite treatment. Finally, the converted DNA fragments were PCR-amplified and sequenced using Illumina HiSeqTM 2500 by Gene Denovo Biotechnology Co., (Guangzhou, China).

### 4.6. Data Filtering

The raw data obtained by sequencing were converted into sequence data by base calling, which we call raw data or raw reads, and the results were stored in the fastq file format. To obtain high-quality clean reads, raw reads were filtered according to the following rules: (1) removing reads containing more than 10% of unknown nucleotides (N); (2) removing low-quality reads containing more than 40% low-quality (Q-value ≤ 20) bases. The correlation values of the three biological replicates for each sample calculated for NQs vs. CQs and are presented in Appendix A.

### 4.7. Mapping Reads to the Reference Genome

The obtained clean reads from each sample were mapped to the species reference *A. cerana* genome (ASM1110058v1) using BSMAP software [48]. Then, a custom Perl script was used to call methylated cytosines, which were tested with the correction algorithm described in [54]. The sequencing results were compared with the reference genome, and only the sequences in the unique positions on the alignment were used for subsequent standard information analysis and personalized analysis. The methylation level was calculated based on the percentage of methylated cytosine in the whole genome, in each chromosome, and in different regions of the genome for each sequence context (CG, CHG and CHH) [55]. To assess different methylation patterns in different genomic regions, the methylation profile was plotted for flanking 2 kb regions and gene body (or transposable elements), based on the average methylation levels for each window.

### 4.8. Bisulfite Treatment Efficiency Detection

The efficiency of bisulfite processing greatly affects the determination of methylation sites. In the experiment, lambda DNA (lambda phage DNA) was generally added to the DNA of the sample being tested (usually higher animals and plants), and the methylation rate of the detected lambda DNA was calculated to evaluate the efficiency of experimental transformation. Theoretically, all Cs on lambda DNA are unmethylated, and after bisulfite treatment, all C will theoretically be converted to U (that is, T in sequencing), but sites where the transformation fails will remain unchanged. The statistical sequence determined that the proportion of C sites converted to T sites in lambda DNA sequencing results is equivalent to the efficiency of the bisulfite experiment. In general, bisulfite treatment with a conversion rate of more than 99% is considered to be reliable using bisulfite processing with ZYMO EZ DNA Methylation-Gold kit, according to [56]. The conversion rate of each sample is shown in Appendix A: the first column is the sample name, and the second column is the conversion rate.

### 4.9. Differential Methylation Analysis

We used methylkit (V1.4.1) software [57] for differential DNA methylation analysis. We first filtered to remove the low sequencing depth site. The C site requires a minimum sequencing depth greater than or equal to 4 for subsequent differential methylation analysis. We used a 200 bp window to scan the whole genome, calculated the average DNA methylation rate within each window (some type of C), and then compared the differences in methylation levels within each window of each sample. For all C, CG, CHG, and CHH, we performed differential methylation analysis separately and filtered using different standards.

To identify differentially methylated regions (DMRs) between two samples, the minimum read coverage to call a methylation status for a base was set to 4. DMRs for each sequence context (CG, CHG and CHH) were determined according to the following criteria: (1) For CG, the numbers of GC in each window should be ≥5, the absolute value of the difference in methylation ratio should be ≥0.25, and *p* ≤ 0.05. (2) For CHG, the numbers in a window should be ≥5, the absolute value of the difference in methylation ratio should be ≥0.25, and *p* ≤ 0.05. (3) For CHH, the numbers in a window should be ≥15, the absolute value of the difference in methylation ratio should be ≥0.15, and *p* ≤ 0.05; (4) For all C, numbers in a window should be ≥20, the absolute value of the difference in methylation ratio should be ≥0.2, and *p* ≤ 0.05. The DMRs were mapped to honey bee official genes to identify differentially methylated genes (DMGs).

### 4.10. Functional Enrichment Analysis of DMGs

To analyze the functional enrichment of genes affected by DMRs, Gene Ontology (GO) and KEGG pathway enrichment analyses were conducted using DMGs. GO enrichment analysis provides all GO terms that are significantly enriched in genes compared to the genome background and filters the genes that correspond to biological functions. Firstly, all ceRNAs were mapped to GO terms in the Gene Ontology database (http://www.geneontology.org/, accessed on 15 June 2022), and gene numbers were calculated for every term. Significantly enriched GO terms in genes compared to the genome background were defined using a hypergeometric test [58,59].

Genes usually interact with each other to play roles in certain biological functions. Pathway-based analysis helps to further understand the biological functions of genes [60]. KEGG is the major public pathway-related database (http://www.kegg.jp/kegg/, accessed on 15 June 2022). Pathway enrichment analysis significantly identified enriched metabolic pathways or signal transduction pathways in genes compared with the whole genome background. For KEGG enrichment analysis, the top 20 KEGG terms were selected. To test the statistical significance of the enrichment of DMGs in KEGG pathways (Q-value < 0.05), we used the KOBAS 2.0 software [61,62,63].

## Figures and Tables

**Figure 1 ijms-25-03368-f001:**
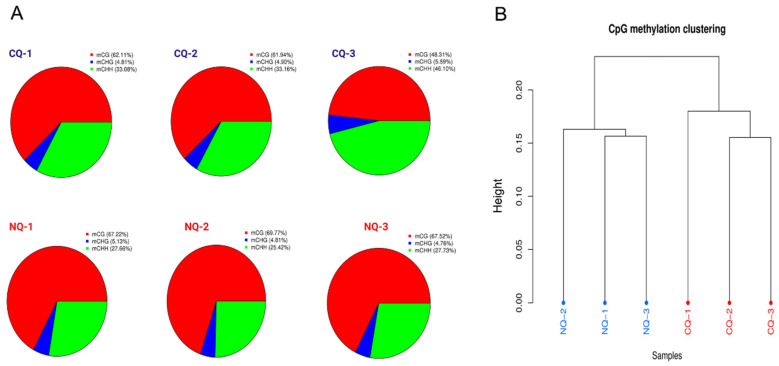
(**A**) The number of methylated C bases of different sequence types. Pie charts demonstrate the proportions of mCG, mCHG, and mCHH between NQs and CQs, with mCG being the most abundant, ranging from 48.31 to 69.77%. The NQs have a higher CG% than control queens. (**B**) The cluster analysis between samples based on methylation rate information from CG base sites. Dendrogram clearly showing that NQs are separated from different CQs. The CPG methylation clustering tree exhibits distinct differences between samples of NQs marked in blue and CQs marked in red.

**Figure 2 ijms-25-03368-f002:**
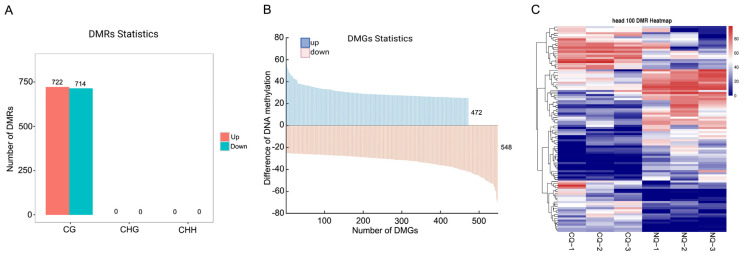
(**A**) Number of DMRs between NQs and CQs. Bar plots show the total number of DMRs of three mCG types between NQs and CQs. Red bars are the number of up-methylated DMRs in NQs compared to CQs, and green bars are the down-methylated DMRs (**B**) DMR-related DMGs. Upward cyan bars are the methylation levels of DMGs in NQs, whereas downward orange bars are the methylation levels of DMGs in CQs. (**C**) Clustering of 100 DMRs between NQs and CQs. Each group has three biological replicates.

**Figure 3 ijms-25-03368-f003:**
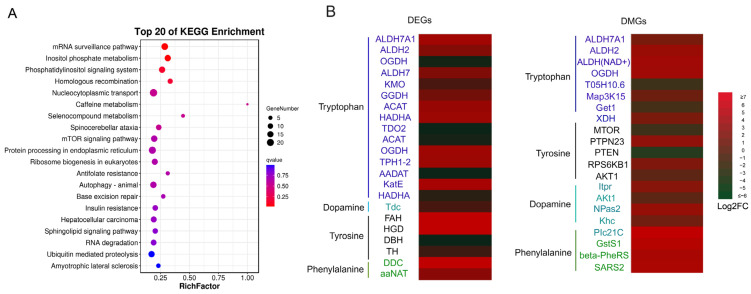
(**A**) The top 20 pathways of KEGG enrichment analysis of DMGs between NQs and CQs. The sizes of circles represent the number of DMGs, and the colors of circles represent the *p*-values of enrichment. (**B**) The heatmap of key DMGs and DEGs that are related to honey bee body color alternation in nutritional crossbreeds. These key DMGs and DEGs were enriched in four key KEGG pathways including the tyrosine, tryptophan, dopamine, and phenylalanine pathways. The DEGs are from our previous study [12]. Different colors represent the significantly differential expression of key DMGs, and DEGs using their log2 (fold change) values.

**Figure 4 ijms-25-03368-f004:**
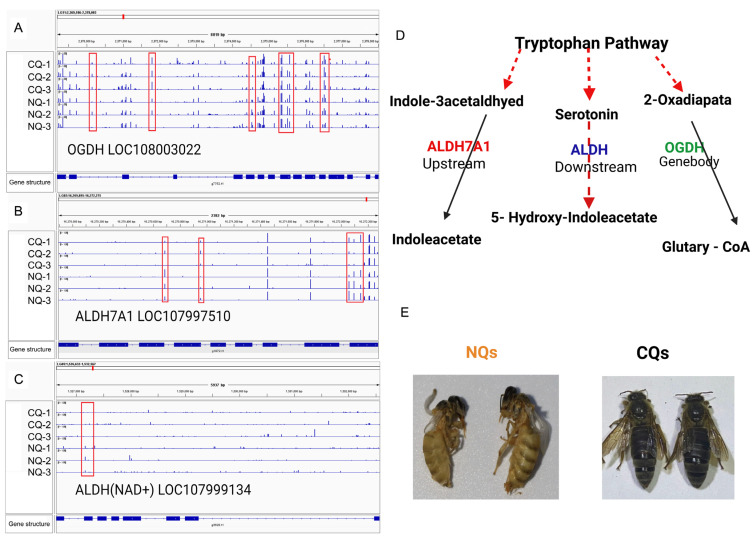
(**A**–**C**) The tryptophan pathway and key genes. The DNA methylation levels of each CpG island of three key DMGs [(**A**) *OGDH*, LOC108003022; (**B**) *ALDH7A* LOC107997510; (**C**) *ALDH(NAD+)* LOC107999134] in NQs and CQs. The different CpG sites between NQs and CQs are marked with red boxes. The top lines of panel A, B and C are the gene locations in honey bee genome. The middle blue vertical bars are the methylation levels of each CpG site in six samples (CQ-1, CQ-2, CQ-3, NQ-1, NQ-2 and NQ-3), and the bottom lines are their gene structures with bold blue lines as exons and thin lines as introns. (**D**) The tryptophan KEGG pathway includes three differentially methylated candidate genes related to regulating body color. The dashed red arrows are predicted pathways in KEGG and the black arrows are verified pathways in KEGG. (**E**) The body color of NQs and CQs in this study.

**Table 1 ijms-25-03368-t001:** Summary of genome-wide methylation sequencing data.

	Before Filter				After Filter			
Samples	Clean Data (bp)	Q20%	N%	GC%	Clean Data (bp)	Q20%	N%	GC%
CQ-1	6,677,533,800	6,004,094,898 (89.91%)	68,826 (0.0%)	994,113,398 (14.88%)	6,579,118,528 (98.53%)	5,934,878,425 (90.21%)	67,815 (0.0%)	939,863,645 (14.28%)
CQ-2	7,590,253,800	6,973,265,799 (91.87%)	97,377 (0.0%)	1,088,876,017 (14.35%)	7,533,897,309 (99.26%)	6,935,328,057 (92.05%)	96,666 (0.0%)	1,068,174,594 (14.18%)
CQ-3	7,431,298,800	6,773,456,838 (91.15%)	96,805 (0.0%)	1,154,870,784 (15.54%)	7,360,162,773 (99.04%)	6,726,348,781 (91.39%)	95,856 (0.0%)	1,128,746,356 (15.34%)
NQ-1	8,955,201,900	8,178,295,710 (91.32%)	97,416 (0.0%)	1,375,753,231 (15.36%)	8,862,330,769 (98.96%)	8,114,049,174 (91.56%)	92,520 (0.0%)	1,332,182,306 (15.03%)
NQ-2	6,638,377,200	6,037,648,753 (90.95%)	73,693 (0.0%)	1,025,061,496 (15.44%)	6,573,340,409 (99.02%)	5,992,949,234 (91.17%)	70,632 (0.0%)	995,135,431 (15.14%)
NQ-3	7,353,264,300	6,695,053,336 (91.05%)	81,142 (0.0%)	1,040,993,533 (14.16%)	7,298,867,089 (99.26%)	6,658,844,959 (91.23%)	78,310 (0.0%)	1,020,328,299 (13.98%)

## Data Availability

The raw bisulfite data from this study can be found in NCBI database under the accession: NCBI Bioproject PRJNA2757128.

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
