# Peer review of "Roles of DNA Methylation in Color Alternation of Eastern Honey Bees (Apis cerana) Induced by the Royal Jelly of Western Honey Bees (Apis mellifera)"

_ijms, 2024, doi:10.3390/ijms25063368_

Round 1

Reviewer 1 Report

Comments and Suggestions for Authors

The document presents a robust study on the effects of DNA methylation on color alteration in honey bees fed with cross-species royal jelly, offering significant insights into epigenetic mechanisms. The study is meticulously conducted, with detailed methods and conclusive findings that contribute to the understanding of phenotypic plasticity in insects. It combines genome-wide methylation analysis, differential methylation analysis, and functional enrichment analysis to elucidate the molecular underpinnings of color change. The results are clearly presented, supported by relevant data, and discussed in the context of existing literature, highlighting the novelty and importance of the findings. The manuscript was changed according to previous remarks and in my opinion is ready to publish.

Author Response

Response to Reviewer 1

Comments and Suggestions for Authors

The document presents a robust study on the effects of DNA methylation on color alteration in honey bees fed with cross-species royal jelly, offering significant insights into epigenetic mechanisms. The study is meticulously conducted, with detailed methods and conclusive findings that contribute to the understanding of phenotypic plasticity in insects. It combines genome-wide methylation analysis, differential methylation analysis, and functional enrichment analysis to elucidate the molecular underpinnings of color change. The results are clearly presented, supported by relevant data, and discussed in the context of existing literature, highlighting the novelty and importance of the findings. The manuscript was changed according to previous remarks and in my opinion is ready to publish.

Thank you so much for your revision and for your valuable comments.

Reviewer 2 Report

Comments and Suggestions for Authors

Dear Authors,

The presented article is methodologically and substantively advanced, and the research goal has been clearly achieved.

Most graphic figures correspond with the result presentation.

I localized a few ambiguities for the reader in this manuscript:

 Lines 162-164: The presentation of the result is unclear. It isn't easy to compare with the data in Figure S4.

Moreover, all Tables S (S1-S7) were cited in the text but were not visible in the supplementary materials, making it difficult to interpret the results.

Line 322 should be native.

Section 4.1 Please state clearly how many colonies of each species were used to study (as a source of eggs?) and how many colonies were used to obtain royal jelly.

Sections 4.1-4.2 – perhaps some scheme for the whole procedure would be useful, just for proper understanding by a broader readership?  Why were other ingredients - sugars - added to royal jelly.?

Author Response

Response to Reviewer 2

Comments and Suggestions for Authors:

Dear Authors,

       The presented article is methodologically and substantively advanced, and the research goal has been clearly achieved. Most graphic figures correspond with the result presentation. I localized a few ambiguities for the reader in this manuscript:

1-Lines 162-164: The presentation of the result is unclear. It isn't easy to compare with the data in Figure S4.

Response 2 Thanks for your helpful comment. You’re right that it is not easy to compare the results in Fig.S4. Here we have added a new excel table which contains the detailed GO enrichment results, please see table S8.

2-Moreover, all Tables S (S1-S7) were cited in the text but were not visible in the supplementary materials, making it difficult to interpret the results.

Response 3 Thanks for your helpful comment. We already uploaded the supplementary tables, and here we have uploaded all supplementary materials again.    

3-Line 322 should be native.

Response 4 Thanks for your helpful comment. We have deleted this incorrect word and rewritten the sentence in line 276.

4-Section 4.1 Please state clearly how many colonies of each species were used to study (as a source of eggs?) and how many colonies were used to obtain royal jelly.

Response 5 Thanks for your helpful comment. We have re-written this paragraph as follow: Six healthy A. cerana colonies were used as one-day-old larvae suppliers to artificially rear nutritional crossbreed queens (NQs), and same larvae were reared in three of these six colonies to produce natural queens as a control (CQs). Each A. cerana colony had a mated queen and around 12,000 worker bees.  Three strong A. mellifera colonies were used for fresh royal jelly collection, and each colony had a mated queen and more than 30,000 worker bees. Totally six A. cerana colonies and three A. mellifera colonies were used in this study. (Please see line 275-280)

5-Sections 4.1-4.2 – perhaps some scheme for the whole procedure would be useful, just for proper understanding by a broader readership?  Why were other ingredients - sugars - added to royal jelly.?

Response 6 Thanks for your helpful comment.

Artificially rearing the honeybee larvae has been long-termly studied. Only using fresh royal jelly to rear bee larvae in cultural cells in an incubator cannot be succeed, since sugars are vital ingredients for larvae queen development as well yeast extract which provides larvae with protein (Kaftanoglu et al., 2011; Crailsheim et al., 2013). Without these conponents, it is impossble to artificially rear honeybee larvae under a lab condition.Therefore, we used the standardized diet formula and rearing method according to (Schmehl et al., 2016; Zhu et al., 2017; and Aupinel et al., 2005). In addition, we have artificially obtained the nutritional crossbreed A. cerana queens (NQs) sufficiently using the same diet formula Abdelmawla et al., (2023). Thus this technology is quite mature. Here we have re-written this section, please see line 283-297.

References

Schmehl, D.R.; Tomé, H.V.V.; Mortensen, A.N.; Martins, G.F.; Ellis, J.D. Protocolo Para La Cría in Vitro de Obreras de Apis Mellifera. J. Apic. Res. 2016, 55, 113–129, doi:10.1080/00218839.2016.1203530.

Zhu, K.; Liu, M.; Fu, Z.; Zhou, Z.; Kong, Y.; Liang, H.; Lin, Z.; Luo, J.; Zheng, H.; Wan, P.; et al. Plant MicroRNAs in Larval Food Regulate Honeybee Caste Development. PLoS Genet. 2017, 13.

Aupinel, P.; Fortini, D.; Dufour, H.; Tasei, J.; Michaud, B.; Odoux, J.; & Pham-Delegue, M.. Improvement of artificial feeding in a standard in vitro method for rearing Apis mellifera larvae. Bulletin of insectology. (2005), 58(2), 107.

Kaftanoglu, O.; Linksvayer, T. A., & Page Jr, R. E.. Rearing honey bees, Apis mellifera, in vitro I: effects of sugar concentrations on survival and development. Journal of Insect Science. (2011), 11(1), 96.

Crailsheim, K. ; Brodschneider, R. ; Aupinel, P. ; Behrens, D. ; Genersch, E. ; Vollmann, J. ; & Riessberger-Gallé, U.. Standard methods for artificial rearing of Apis mellifera larvae. Journal of Apicultural Research. (2013), 52(1), 1-16.

Abdelmawla, A.; Yang, C.; Li, X.; Li, M.; Li, C.L.; Liu, Y.B.; He, X.J.; Zeng, Z.J. Feeding Asian Honeybee Queens with European Honeybee Royal Jelly Alters Body Color and Expression of Related Coding and Non-Coding RNAs. Front. Physiol. 2023, 14, 1–12, doi:10.3389/fphys.2023.1073625.
